# The Eco-Bio-Social Factors That Modulate *Aedes aegypti* Abundance in South Texas Border Communities

**DOI:** 10.3390/insects12020183

**Published:** 2021-02-21

**Authors:** Jose G. Juarez, Selene M. Garcia-Luna, Matthew C. I. Medeiros, Katherine L. Dickinson, Monica K. Borucki, Matthias Frank, Ismael Badillo-Vargas, Luis F. Chaves, Gabriel L. Hamer

**Affiliations:** 1Department of Entomology, Texas A&M University, College Station, TX 77843, USA; selene.marysol@gmail.com (S.M.G.-L.); ibadill1@southtexascollege.edu (I.B.-V.); 2Pacific Biosciences Research Center, University of Hawaii at Mānoa, Honolulu, HI 96822, USA; mcmedeir@hawaii.edu; 3Colorado School of Public Health, Department of Environmental and Occupational Health, Aurora, CO 80045, USA; katherine.dickinson@cuanschutz.edu; 4Lawrence Livermore National Laboratory, Livermore, CA 94550, USA; borucki2@llnl.gov (M.K.B.); frank1@llnl.gov (M.F.); 5Instituto Costarricense de Investigación y Enseñanza en Nutrición y Salud (INCIENSA), Tres Ríos 4-2250, Cartago, Costa Rica; lfchavs@gmail.com

**Keywords:** *Aedes aegypti*, Knowledge Attitudes and Practices (KAP), risk factors, vector control, integrated vector management

## Abstract

**Simple Summary:**

The *Aedes aegypti* mosquito is distributed worldwide and has become a major public health concern due to its proclivity for the urban environment, human feeding behavior, and ability to transmit agents of diseases such as Zika, chikungunya, and dengue. In the continental United States, the region known as the Lower Rio Grande Valley is one of the few areas with local mosquito transmission of these pathogens transmitted by *Ae. aegypti*. With limited resources for mosquito control in this region, understanding the ecological, biological, and social factors that affect *Ae. aegypti* population can help guide and improve current control efforts. We were able to observe widespread knowledge regarding Zika, but with very low importance given to mosquitoes as a problem. We found that the presence of window-mounted air conditioning units, number of windows and doors, characteristics of the property, and presence of children in the household all influenced the abundance of *Ae. aegypti*. The current results not only show a need for improved community engagement for increasing disease and mosquito risk awareness, but also provide risk factors that can guide current vector control activities.

**Abstract:**

*Aedes aegypti* control requires dedicated resources that are usually scarce, limiting the reach and sustainability of vector control programs. This generates a need to focus on areas at risk of disease transmission and also understand the factors that might modulate local mosquito abundance. We evaluated the eco-bio-social factors that modulate indoor and outdoor relative abundance of female *Ae. aegypti* in communities of South Texas. We conducted housing quality and Knowledge Attitudes and Practices surveys in households that were part of a weekly mosquito surveillance program in November of 2017 and 2018. Our results showed widespread knowledge of mosquitoes and Zika virus by our participants. However, less than 35% considered them as serious problems in this region. The presence of window-mounted air conditioning units increased the risk of female mosquito relative abundance indoors. An increase in outdoor relative abundance was associated with larger properties and a higher number of children between 6 to 17 years of age. Interestingly, we observed that an increasing number of children <5 years of age modulated both indoor and outdoor relative abundance, with a 52% increase indoors and 30% decrease outdoors. The low perception of mosquito and disease risk highlights engagement needs for vector-borne disease prevention in this region. The identified risk factors can help guide public health officials in their efforts to reduce human and vector contact.

## 1. Introduction

The yellow fever mosquito, *Aedes* (*Stegomyia*) *aegypti* (L.), is the main vector of several arboviral pathogens that place more than half a billion people at risk of infection per year globally [1,2,3]. With recent epidemics of dengue, chikungunya, and Zika in the Americas, *Ae. aegypti* remains as a major public health concern due to its highly anthropophilic behavior and affinity to man-made container habitats [4,5,6,7]. Due to the absence of effective vaccines, efforts to control the spread of these arboviruses continue to target *Ae. aegypti* [3]. However, current vector control programs require dedicated resources, that are scarce, which prevent the expansion and sustainability of such programs [8]. This creates a constant need to focus on regions that are at the highest risk for disease transmission and to develop knowledge of local risk factors that might help reduce the burden of these vector-borne disease.

Prior work has shown that domestic water-holding containers [9], socio-demographic characteristics [10], and ecological factors such as climate and altitude [8,11] are associated with the abundance of *Aedes* spp. mosquitoes. Such information has helped to develop guidelines and new entomological indices for the surveillance and control of these species [12]. An in-depth understanding of fine-scale ecological, biological, and social factors that modulate *Ae. aegypti* populations is needed to help guide vector control activities and assure long term sustainability of programs [13]. In the continental United States (US), *Ae. aegypti* has been recorded in over 220 counties in 28 states with widespread distributions in California, Arizona, Louisiana, Florida, and Texas [14,15]. Despite this widespread distribution only a handful of counties have reported autochthonous transmission of dengue, chikungunya, or Zika viruses [16]. One of the areas at highest risk of *Ae. aegypti* vector-borne disease transmission is the region of the Lower Rio Grande Valley (LRGV) in South Texas, along the US–Mexico border [17]. Several studies in this region have described the seasonal occurrence and abundance of *Ae. aegypti* [17,18], while others have suggested the potential impact of climate change on disease transmission [19] and habitat selection [20], and some have evaluated biological control methods for *Aedes* spp. mosquitoes [21]. However, research on surveillance and control efforts focusing on social-economic factors has been scarce [22,23]. 

Gaining in-depth knowledge of which ecological, biological, and social (eco-bio-social) factors modulate the relative abundance of mosquito populations can provide key information on how to approach different control efforts in a more efficient way [24]. In the contiguous US factors such as household demographics, housing characteristics and peridomicile environment have been shown to modulate *Ae. aegypti* populations [24,25]. Our study focuses on identifying the eco-bio-social factors at the household-level that modulate the indoor and outdoor abundance of female *Ae. aegypti* in the LRGV. We hypothesize that an increase in household occupants will have an increase in the abundance of female *Ae. aegypti*. Inversely, we hypothesize a decrease in the quality of housing characteristics and management of the peridomicile will increase the abundance female *Ae. aegypti*. In the LRGV, improving surveillance efforts and disease awareness of mosquito borne viruses are needed to improve testing and identification of cases [26]. We assessed community members’ knowledge, attitude and practices (KAP) around mosquitoes and the diseases they transmit, allowing us to identify perception gaps that can help guide new community engagement tools to enhance the effectiveness of mosquito control efforts and disease awareness in areas like the LRGV, where disease risk is high, and resources are limited. 

## 2. Materials and Methods

### 2.1. Study Area

The study took place in the counties of Hidalgo (26°06′10.91″ N, 98°15′16.25″ W) and Cameron (26°09′49.69″ N, 97°49′26.36″ W), which are part of the region known as LRGV in South Texas, US (Figure 1). There are an estimated 1.3 million inhabitants within these counties, of which 90% considered themselves of Hispanic or Latin origin, 85% speak Spanish and at least 29% live in poverty [27,28]. Within these counties, there are several un-incorporated communities called ‘*colonias*’, usually inhabited by families of Hispanic heritage who often live in low-income housing and lack city services such as waste management, paved roads, and potable water [29,30,31]. The climate in this region is considered humid sub-tropical, with a cold/dry season from November to February (7–21 °C), and a rainy season that starts in April (18–30 °C), peaks in September (23–33 °C), and finishes in October (19–31 °C) [32].

### 2.2. Community Selection and Sample Size

Communities were selected based on average income level per household, total number of households within the community, isolation of the community, and distance to our base of operation in Weslaco, Texas. Communities were classified into census block groups based on mean household income for the 2010 census (low- ($15,000–$29,999) and middle-income ($30,000–$40,000)) [17]. Initially, we identified nineteen candidate communities of which eight were selected based on security and community participation. These eight communities were grouped into low- (Balli, Cameron, Chapa and Mesquite) and middle-income (Christina Ct., Rio Rico, La Vista and La Bonita) (Table 1). Household recruitment and selection for the weekly surveillance has been detailed by Martin et al. [17]. Briefly, random households were visited and if agreed by the homeowner an indoor and outdoor Autocidal Gravid Ovitrap (AGO) was deployed. We visited enough households until a coverage of one AGO per 100 m^2^ was achieved for each community. If a household dropped out of the study, we tried to recruit its neighbor to the right until a new household was recruited. 

### 2.3. Entomological Surveillance

This study was part of a cluster randomized cross-over trial that focused on evaluating the BioCare Autocidal Gravid Ovitrap (AGO) (SpringStar Inc.) (Figure 2A) as a surveillance and control tool for *Ae. aegypti* in South Texas. Mosquito surveillance was described by Martin et al. [17]. We conducted weekly surveillance of indoor (Figure 2B) and outdoor (Figure 2D) mosquitoes from September 2016 to December 2018. Independently of homeowner presence, outdoor AGOs were surveyed when accessible during the surveys. On each visit, mosquitoes were removed from the glue board (OviCatch, Catchmaster, USA) with a teasing needle and separated in the field by species (*Ae. aegypti*, *Ae. albopictus*, *Culex* spp., and other spp.), sex (male and female), and female condition (unfed, gravid and blood fed) (Figure 2C). Glue boards with more than 35 mosquitoes were replaced with a new one. Mosquitoes on the removed glue boards were identified at our laboratory in Weslaco, Texas using taxonomic keys [33,34]. During each weekly visit, we changed the hay infusion (~ 3.5 L of water and 3 g of hay) of the indoor and outdoor AGO, and glue boards were replaced as needed (usually every two months [35]) (Figure 2E). The AGO intervention was carried out during the months of August through December of both 2017 and 2018. 

### 2.4. KAP and House Quality Surveys

Households were surveyed between November 8th–22nd in 2017 (*n* = 33) and 2018 (*n* = 4). The surveys done in 2018 are the dropout replacements of 2017. We used a structured face-to-face questionnaire to characterize the knowledge, attitudes, and practices (KAP) of household adults towards mosquitoes and the diseases they transmit. The questionnaire consisted of close-ended, semiclosed-ended, and ranking items related to household demographics, mosquitoes, Zika virus, and vector control [36,37]. We also conducted a house/peridomicile environmental survey to assess house construction materials and the presence of peridomestic containers. Houses were evaluated counterclockwise from the main house entrance. We recorded housing materials (timber/metal, cement, brick), screen quality (with holes, with no holes, size of holes) on windows and doors, the type of air conditioning (A/C) unit (window mounted, central) if present, and the type and number of mosquito container habitats found in each household peridomicile (Appendix A). We defined the peridomicile of a household as the area found from the property limit to the main house perimeter. 

A random selection of six households that were not included in the AGO surveillance efforts were used for field validation of the KAP and house/peridomicile surveys prior to their implementation. Each survey lasted approximately 20 min and was carried out by two team members. The surveys were developed in English and Spanish, with both versions validated by an external bilingual reviewer familiar with the colloquial language of the LRGV to assess consistency (Appendix A). 

### 2.5. Statistical Analysis

Our main outcome variable of interest was indoor and outdoor relative abundance of female *Ae. aegypti*. To construct these outcome variables from our data, we focused on a reduced temporal dataset. This dataset only included the surveillance efforts carried out from January 8th (week 2) to August 8th (week 32) of 2018 (households = 32; average weeks of surveillance = 18, SD = 6.8) (Appendix A). This was done to account for the effect of AGO coverage (measured as the number of traps and houses within an area based on the dispersal of female *Ae.* aegypti [38]) during the intervention periods (August to December), since it has been observed that these traps modulate the abundance of female *Ae. aegypti* [39]. This reduced dataset provided us with a surveillance period that was balanced, consistent, and comparable between households and communities. 

To quantify the association between the eco-bio-social factors with the abundance of indoor and outdoor female *Ae. aegypti,* we used generalized linear models (GLM) for count data. Initially, we determined the error distribution by modeling the mosquito count data using a Poisson distribution (variance = mean) and then compared it with those of overdispersed counts with a negative binomial distribution (variance > mean) [40,41]. We used a negative binomial type 1 and type 2 distributions to evaluate if the variance increased linearly or quadratically with the mean [42]. This was done to correctly specify the variance-mean relationship of the models, since this can affect the weighted least-squares algorithm during the fitting of the data. This variance–mean relationship might produce drastic differences when estimating the abundance of mosquitoes because small and large counts are weighted differently for the negative binomial type 2 distribution [43,44]. Models were fitted using a log link function and the Laplace method in the “glmmTMB” package in R 3.5.1 (R Core Team, Vienna, Austria) [45,46,47]. This framework was chosen given its ability to account for the unbalanced nature of our dataset, due to variation on the sampling effort, i.e., the number of weeks of trapping, since we depended on the homeowner’s presence to check the AGOs. We employed the offset function with the term weeks of trapping to account for differences in sampling effort that was constrained by our ability to access ovitraps and collect mosquitoes. The residuals of the models were plotted to evaluate the error heteroscedasticity, qq-plots, and leverage points. We also evaluated the spatial independence of the residuals using the Moran’s I test [48]. 

Fixed effects were selected from the indices developed using the KAP and housing quality surveys. To reduce the large number of variables in both surveys (KAP: 73 variables and housing quality: 55 variables) we used dimension reduction methods (DRM) which generate five indices (AP1, AP2, yard, window, and door) [49,50]. Firstly, we performed descriptive statistics on the KAP and housing variables to summarize the results obtained and assessed the variables that had a low standard deviation or an extreme frequency (high or low) which would carry a low weight when estimating the principal components (PC) for continuous variables, or the multiple correspondence analysis (MCA) dimensions for categorical variables [51,52]. The AP1 and AP2 indices were generated by either grouping variables that were highly clustered, or collinear with other covariates, respectively. Meanwhile the yard, window and door indices were generated by grouping variables that were similar in nature (i.e., total windows, window screen, window screen holes, etc.). The tradeoff of grouping variables using DRM is that for each component or dimension not retained, we lose a proportion of the variation among the original variables which could lead to the loss of relevant information. Moreover, evaluating the statistical significance of the original variables is not possible. DRM were estimated using the correlation matrix of the considered variables to avoid estimation problems associated with variances of different magnitude, using the “FactoMineR” package in R [49,53]. The AP1 index was estimated using multiple correspondence analysis (MCA) since we only had categorical variables, and the yard index was estimated using factor analysis of mixed data (FAMD), since we had both quantitative and qualitative variables. Principal component analysis (PCA) was used for the indices that only had continuous variables (Appendix A). No analysis was conducted on the knowledge of participants regarding mosquitoes and their diseases, since the survey was conducted during an active mosquito intervention. We only selected indices that explained >50% of the cumulative variability in the first two PC (or MCA and FAMD dimensions) as fixed effects in the GLM. Variables that showed collinearity during the DMR but were not grouped in an index were evaluated using variance inflation factor (VIF) of less than five to select as fixed effects in the GLM [54]. Variables with the highest VIF, at each step, were systematically dropped until no VIF was above 5. For a detailed description on how the indices were developed please refer to Appendix A.

The indoor models evaluated had fixed effects for type of AC unit (3 levels), opening window for ventilation (2 levels), opening door for ventilation (2 levels), water storage (2 levels), and income (3 levels: <$25k; $25–50k and >$75k, no interviewee reported earning between $50–75k), with covariate effects for other containers (all other water holding containers that were not tires), outdoor female *Ae. aegypti*, the second PC of the door index and the first two PC of the window index, and the first two dimensions of the AP2 index. The outdoor models evaluated had fixed effects for shade vegetation (4 levels), messy yard (3 levels: orderly = grass < 5cm + debris organized in apparent order + no trash; average = only two of previous criteria, disorderly = none of previous criteria) and water storage (2 levels), with covariate effects for other containers, tires and the first two PC of window and door indices and the first two dimensions of the AP2 index (Table 2). Models were simplified using backward elimination [55], where single parameters were sequentially removed based on the significance (*p* < 0.05) of the fixed effects estimates. The best fit model was selected based on minimizing Akaike information criterion (AIC) [56]. Models and graphs were generated using R 3.6.1 (R Core Team, Vienna, Austria). 

## 3. Results

### 3.1. KAP: Aedes aegypti and Zika

The knowledge of community members regarding mosquitoes was high with 97% recognizing adult mosquitoes. However, fewer respondents were able to identify mosquito larvae (43%). When asked whether they thought mosquitoes were a problem in their community, 85% said they were, with 31% saying they were a serious problem. Most respondents (87%) could name at least one disease transmitted by mosquitoes, of which 79% recognized Zika, 56% dengue, and 6% West Nile virus. It appears that there is widespread knowledge of Zika in this region: 85% of respondents had heard about the disease prior to this survey. However, only 29% considered Zika to be a serious problem in the LRGV, of which 55% said that it was due to their family and children (Table 3). 

### 3.2. KAP: Prevention, Control, and Demographics

Ninety five percent of interviewees confirmed they believed something should be done if they had a mosquito problem on their property. Of those, 78% considered the use of insect repellents as a method for control. We observed that 14% of interviewees considered calling the city or the county for mosquito control efforts; interestingly, all belonged to a middle-income community. With regard to supporting an AGO intervention in their communities, 95% said they would if the three traps and maintenance were given without a cost (Table 4). Household demographics showed that 31% had toddlers (children <5 years of age), with 84% of houses having between 1–6 residents. In addition, more than half of all houses (63%) reported earning < $24,999 (Appendix A). The AP1 index was not used since it only account for a cumulative variation of 31.3% for the variables used. The AP2.1 index can be viewed as a measure of the lack of children ≤5 years of age in a house. The AP2.2 index can be viewed as a measure of larger property size and a greater number of children of 6 to 17 years of age in a house (Appendix A). The AP2 index had a cumulative variation of 55.2%.

### 3.3. Housing Materials: Yard, Windows, and Doors

We observed that all houses surveyed had a patio with grass in their premises. Lots within our communities averaged 770 m^2^ (SD = 220m^2^). When surveying the peridomicile we found that the most common container that could hold water was for drainage plates of plant pots with 90% houses having at least one, followed by tin cans 46%, tires 44%, and drum water barrels 5%. We also observed that only one house did not have an A/C unit, while 54% had a central system and 43% had a window mounted unit. Houses with a window AC had an average of 1.5 (SD = 1.7) units per house. These window AC units were capable of cooling only the room where they were located. We also observed that 88% of the doors had immediate access to the exterior premises of the house, with an average of 2.3 (SD = 0.9) exterior doors per house (Table 5). The Yard index was not used, since it only account for a cumulative variation of 35.1% for the variables used (Appendix A). The Window1 index can be viewed as a measure of quantity of windows, while Window2 captures quality (higher values denote poorer quality). The window index had a cumulative variation of 50.6%. The Door1 index can be viewed as a measure of quality (higher values denote poorer quality), while Door2 is a measurement of quantity of doors with an exterior access (Appendix A). The door index had a cumulative variation of 62.2%.

### 3.4. Factors Associated with Indoor and Outdoor Relative Ae. aegypti Abundance

Indoor abundance: We determined no spatial autocorrelation with an observed Moran’s I test of 0.06 (expected = −0.03, SD = 0.12, *p* = 0.4) and that the best distribution for our dataset was a negative binomial type 2, since the Poisson model showed overdispersion. The best fit model (m5) had an AIC of 139.7 (Appendix A). Holding other variables constant, households that had window-mounted AC units had 4.7 (Exponentiated 95% CI: 1.5–15.6) times more indoor female mosquitoes compared to households with a central AC system. Households that reported opening the window for ventilation had 3.7 (Exponentiated 95% CI: 1.2–13.3) times more indoor female mosquitoes compared to households that did not open the window. Households with a higher number of windows and doors with an exterior access had 2.1 (Exponentiated 95% CI: 1.3–3.6) and 1.66 (Exponentiated 95% CI: 1.01–2.9) times more female mosquitoes found indoors respectively. Interestingly, households that had fewer children ≤5 years of age had 0.49 (51%, AP2.1, Exponentiated 95% CI: 0.3–0.8) times fewer indoor female mosquitoes (Table 6).

Outdoor abundance: We determined no spatial autocorrelation with an observed Moran’s I test of 0.12 (expected = −0.03, SD = 0.13, *p* = 0.2). The Poisson distribution residual plot analysis showed overdispersion and we determined that the best distribution was a negative binomial type 1. The best fit model (m13) had an AIC of 325.9 (Appendix A). Holding other variables constant, households that reported having an income $25–50k had 5 (Exponentiated 95% CI: 2.7–9.3) times more outdoor female mosquitoes compared to households with an income <$25,000. Higher abundance of outdoor female mosquitoes was also observed for households with fewer children ≤5 years of age (1.7, AP2.1, Exponentiated 95% CI: 1.3–2.1), larger properties with more children between 6–17 years of age (1.4, AP2.2, Exponentiated 95% CI: 1.1–1.9) and households with poor door quality (1.2, Door1, Exponentiated 95% CI: 1.1–1.4). Interestingly, households that had more tires in their properties had 8% (0.92, Exponentiated 95% CI: 0.8–0.9) fewer female mosquitoes. Lower abundance of female mosquitoes was also observed for households that had yards with >51% covered grass (0.38, Exponentiated 95% CI: 0.2–0.7), doors that were open more often (0.30, Exponentiated 95% CI: 0.2–0.5) and had more doors with outdoor access (0.7, Door2, Exponentiated 95% CI: 0.6–0.8) (Table 7).

## 4. Discussion

We analyzed the eco-bio-social factors that modulate the indoor and outdoor relative abundance of female *Ae. aegypti.* We also present the associated perception of community members regarding mosquitoes, the diseases they transmit and control measures in one of the few areas in the continental US that has vector borne Zika transmission [26]. We were able to observe widespread knowledge within these communities regarding mosquitoes, Zika and its methods of transmission, with over 90% of participants capable of identifying adult mosquitoes and at least one arboviral disease. However, less than 35% of participants considered mosquitoes, and Zika, as serious problems for this region. Of those who acknowledged a problem, 55% said it was due to having small children or a pregnant family member. Interestingly, we were able to observe that households that had more children ≤5 years of age had a 51% increase on the number of female *Ae. aegypti* indoors and a 30% reduction in the number of female *Ae. aegypti* outdoors. This could indicate that households with small children are managing their peridomicile environment more efficiently to reduce *Ae. aegypti* population outdoors, but some of the females available might be brought indoors after the recreational activities small children have in the yard. We believe this might happen if doors are left open or there is back and forth movement by small children from the indoor and outdoor household environment.

Understanding how community members perceive a vector and its associated risk of disease transmission should be a key surveillance component of any vector control program. This allows for the development of culturally appropriate information, establishment of trust with community members, improvements in vector control activities through active community engagement [57,58,59]. The KAP results for mosquito control clearly showed that community members believed mosquitoes should be managed in their property if found. The use of chemical-based methods was the most common form of control, either personal repellents (79%) and/or spraying their properties (35%). We also observed that of the 14% that reported calling either the city or county health authorities for help with mosquito control, all belonged to middle-income communities. When asked about the willingness to support an AGO intervention, we recorded an overwhelming support if traps and maintenance were given free of cost to the participants (95%). However, the support decreased if maintenance had to be provided by the homeowner (63%) and even further if it had an added cost of $15 per trap (25%). It is worth noting that 67% of participants were not sure if other neighbors would support an intervention such as this, which shows the need to understand how willingness to pay might affect novel mosquito control tools sustainability. The results show that most community participants are willing to control mosquitoes, but rely on their own means to handle mosquito populations and, given their limited income, would prefer not to assume the cost. With that in mind, communities within the LRGV might be ideal candidates to evaluate novel control tools if this is provided free or at a lower cost than what we asked and with proper community engagement.

The relative abundance of indoor and outdoor *Ae. aegypti* was modulated by different factors within our communities. Indoor abundance was mostly affected by the presence of window mounted A/C units, with a six-fold increase when compared to households with a central system. The effect of A/C on mosquito abundance is something that has been previously observed in the Continental US. In California, the usage of A/C appears to correlate with a decrease for indoor abundance [24] while, in Arizona and Texas, the presence of A/C correlates with decreasing outdoor abundance [25] and dengue infection [60,61,62], respectively. However, in California, the difference between having a window mounted and central system did not show a significant difference for mosquito abundance [24]. We believe that, in our case, this is due to the poor sealing quality of the window units since over 25% had holes >3 cm. Unsurprisingly, people who reported opening their windows for ventilation had almost a 3-fold increase on female *Ae. aegypti* indoors compared to those who did not. We were also able to observe that reported household income had an impact on the abundance of outdoor female *Ae. aegypti*. Interestingly, households that reported an income $25–50k, had a higher abundance of females detected outdoors when compared to an income <$25k. This counterintuitive result has not been previously reported: usually low-income communities are associated with a higher abundance of mosquitoes outdoors [17,23]. We believe that this is because income at the household level is at best an indirect indicator of factors that might impact mosquito population biology in a community. In the LRGV, some household owners living in low-income communities reported earnings on the higher end of the spectrum, while retired participants living in middle-income communities reported the lower end. Outdoor female abundance was also modulated by property size and the density of children between 6–17 years of age, with a higher abundance for households with larger properties and more children and teenagers. The effect of population density and property size has been associated with an increase of female indoor and outdoor abundance, respectively, in California [24]. Our results further confirm that these variables could be used by vector control programs along the US–Mexico border for identifying areas at high risk of *Ae. aegypti*. Similarly, these results suggest that communities with lower quality housing are candidates to engage in programs aiming to improve housing quality, in accordance with the Sustainable Development Goals (UN global goals to achieve a sustainable future by 2030) and evidence about the association of housing quality improvement and reduced vector-borne disease transmission [63,64,65].

Some of the limitations of our study were that we carried out the surveys in households that had been enrolled in an active surveillance of mosquito populations for several months. This might have biased the results obtained regarding the KAP of mosquitoes and its diseases by making the participants aware of our research objectives during recruitment. To try and control for this issue instead of asking if participants knew about mosquitoes, we showed them samples and images of mosquitoes in their different life stages (larvae, pupae, and adult) and inquired what they thought these organisms might be. In regard to the diseases, only Zika was mentioned during the informed consent process; during the interview process, we emphasized if participants had heard of this disease prior to our study. We interpret the results of the AGO as a control tool acceptance with caution since the survey was conducted during an active AGO intervention in the communities. Additionally, the use of only AGOs, a trap that targets gravid females, limits our interpretation regarding host seeking females for human-vector contact. For such purposes, the inclusion of a trap that also targets host seeking females would be beneficial.

## 5. Conclusions

In the US, one of the few places where vector-borne disease transmission of dengue, chikungunya, and Zika has been recorded is the region known as the Lower Rio Grande Valley (LRGV) in South Texas [66,67], located in the US–Mexico border. There is an urgent need to assess the eco-bio-social factors that modulate risks of emerging human pathogens throughout these communities that are at a higher risk of disease transmission. Our results suggest that the communication methods employed during the Zika epidemic of 2016–2018 by public health officials in the LRGV reached many community members in our study region. Our results also showed local risk factors that might help guide public health officials in their communication efforts to reduce human to vector contact. It appears that future mosquito control campaigns in the LRGV could target households with window-mounted AC units and households with children, to help reduce the local abundance of *Ae. aegypti*. Indeed, prior studies have found that educational campaigns targeting school children not only impacts children, but can have a spillover effect on the adult population [68], and the current study reinforces that the homes with children <5 years of age are also higher priority households for vector control.

## Figures and Tables

**Figure 1 insects-12-00183-f001:**
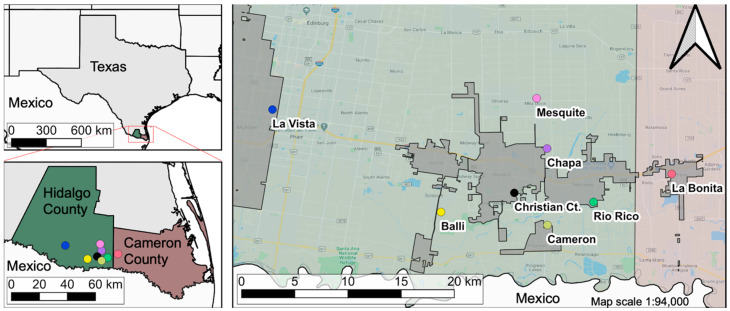
Location of the Lower Rio Grande Valley communities where the Autocidal Gravid Ovitrap surveillance took place. City limits of the communities are shown as a transparent gray area. Only middle-income communities are found within city limits; low-income communities are incorporated into the management area of Hidalgo County. The map was developed using QGIS 3.10 with publicly available administrative boundaries.

**Figure 2 insects-12-00183-f002:**
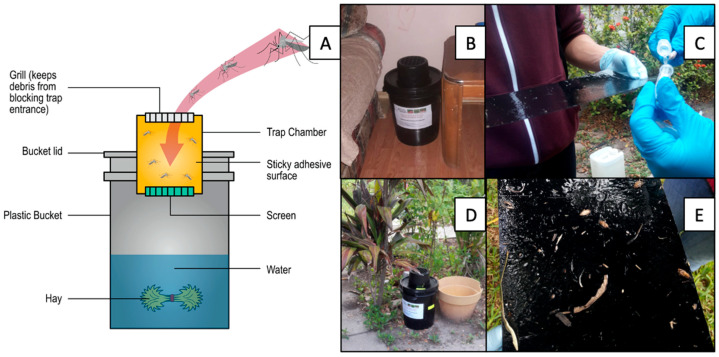
Pictorial diagram and field usage of the AGO trap (Source: CDC, Spring Star, King 5–modified). (**A**) Diagram of the features and components of the AGO trap. (**B**) Indoor placement of AGO. (**C**) Mosquito method of collection from glue board. (**D**) Outdoor placement of AGO. (**E**) Field view of a glue board that needs replacement.

**Table 1 insects-12-00183-t001:** Total number of households in the communities and total number of households with Autocidal Gravid Ovitrap (AGO) and knowledge, attitude and practices (KAP) survey in the Lower Rio Grande Valley.

Income	Community	Total Households	AGO	KAP
Low	Balli	45	7	4
	Cameron	85	6	6
	Chapa	30	5	5
	Mesquite	39	5	5
Middle	Christian Ct.	34	6	5
	Rio Rico	20	5	5
	La Vista	63	6	4
	La Bonita	67	7	6

**Table 2 insects-12-00183-t002:** Generalized linear model (GLM) fixed effect structure, with Akaike information criterion (AIC) distribution analysis.

Target	Offset	Fixed	Distribution (AIC)
Indoor female *Ae. aegypti*	logs (Weeks of Trapping)	TypeAC + OpenWindow + OpenDoor + WaterStorage + OtherContainers + Income + Outdoor female + AP2.1 + AP2.2 + Window1 + Window2 + Door2	Poisson (156.4) Negative Binomial 1 (152.2) Negative Binomial 2 (145.8)
Outdoor female *Ae. aegypti*	logs (Weeks of Trapping)	Vegetation + MessyYard + OpenWindow + OpenDoor + WaterStorage + OtherContainers + Tires + Income + AP2.1 + AP2.2 + Window1 + Window2 + Door1 + Door2	Poisson (804.1) Negative Binomial 1 (337.2) Negative Binomial 2 (340.8)

**Table 3 insects-12-00183-t003:** Knowledge, attitudes, and practices of household owners in the Lower Rio Grande Valley, related to mosquitoes, their diseases, and Zika.

Knowledge, Attitudes and Practices	Response	No. Positive Responses/Total (%)
Mosquitoes and their diseases	Recognized a mosquito larva from picture	17/39 (43.6)
	Recognized an adult mosquito from picture	38/39 (97.4)
	Believed mosquitoes are most abundant during the summer	20/38 (52.6)
	Believed the canals are a source for mosquitoes in their community	17/37 (45.9)
	Had seen a mosquito in the past few days	27/38 (71.1)
	Believed mosquitoes had an impact on their life	33/38 (86.8)
	Health risk	24/33 (72.7)
	Nuisance	12/33 (36.4)
	Considered mosquitoes a problem in their community	33/39 (84.6)
	Small or moderate	21/39 (53.8)
	Serious	12/39 (30.7)
	Knew that mosquitoes can transmit diseases	34/39 (87.2)
	Zika	27/34 (79.4)
	Dengue	19/34 (55.9)
	Chikungunya	6/34 (17.6)
	Malaria	5/34 (14.7)
	West Nile	2/34 (5.6)
	Knew someone that had been infected with dengue, chikungunya and/or Zika	8/39 (20.5)
Zika virus	Had heard about Zika virus before this interview	33/39 (84.6)
	Knew that Zika causes fever symptoms	22/33 (66.7)
	Knew that Zika may affect babies	8/33 (24.2)
	Knew another mode of transmission for Zika besides mosquitoes	13/33 (39.4)
	Sexual intercourse	10/13 (76.9)
	Congenital	1/13 (7.7)
	Considered Zika a problem in the LRGV	22/39 (66.7)
	Somewhat or slightly	14/39 (35.8)
	Very or extreme	8/39 (29.4)
	Worried about Zika because of family and children	12/22 (54.5)

**Table 4 insects-12-00183-t004:** Knowledge, attitudes, and practices of household owners in the Lower Rio Grande Valley, related to prevention and control of mosquitoes.

Knowledge, Attitudes and Practices	Response	No. Positive Responses/Total (%)
Prevention and control of mosquitoes	Had been bitten by mosquitoes inside or outside the home in the past week	22/39 (56.4)
	Stored water on their property for plants and flowers	7/10 (70.0)
	Left windows open for ventilation	19/39 (48.2)
	Left door open for ventilation	17/39 (43.9)
	Believed that they should do something if they had a mosquito problem in their property	37/39 (94.9)
	Use insect repellent	29/37 (78.4)
	Spray insecticide	13/37 (35.1)
	Dump stagnant water	6/37 (16.2)
	Call city or county	5/37 (13.9)
	Limited outdoor activities because of mosquitoes	25/39 (64.1)
AGO intervention	Would support an AGO intervention in their community if the three traps were free and maintenance was provided	37/39 (94.9)
	Would support intervention if AGO traps were free, but household need to provide maintenance	23/37 (62.2)
	Would support intervention if AGO traps were $15 each and household provided maintenance	9/37 (25.0)

**Table 5 insects-12-00183-t005:** Housing and peridomicile variables of the Lower Rio Grande Valley.

Question	Variable	No. Positive Responses/Total (%)
Size of lot (m^2^)	262–600	9/39 (23.1)
	601–1000	19/39 (48.7)
	1001–1204	11/39 (28.2)
No. of bedrooms	1–2	18/39 (46.1)
	3–4	19/39 (48.7)
	5	2/39 (5.1)
Length of vegetation in the yard	Short (< 5 cm)	19/39 (48.7)
	Medium (5–10 cm)	17/39 (43.6)
	Long (>10 cm)	3/39 (7.7)
Houses with containers in peridomicile	Plant pots	35/39 (89.7)
	Tin cans	18/39 (46.2)
	Tires	17/39 (43.6)
	Drum water barrels	2/39 (5.1)
Wall material	Timber/Metal	18/39 (46.1)
	Cement	3/39 (7.7)
	Brick	18/39 (46.2)
Type of roof and material	Flat and cement	2/39 (5.1)
	Pitched and asphalt shingles	37/39 (94.9)
Type of A/C unit	None	1/39 (2.6)
	Window mounted	17/39 (43.6)
	Central system	21/39 (53.9)
Window	With mesh	259/389 (66.6)
	No holes	232/259 (89.2)
	Holes < than 0.5cm	15/259 (5.8)
	Holes ≥ than 0.5cm	13/259 (5.0)
Doors	Exterior door	91/104 (87.5)
	Exterior door with screen	47/91 (51.6)
	Exterior door with gap in the frame	20/91 (21.9)

**Table 6 insects-12-00183-t006:** Main effects statistics for the best fit generalized linear model (NB2) for indoor female *Ae. aegypti* abundance in South Texas.

Variable	Exp (Estimate)	Estimate	Std. Error	95% CI
(Intercept)		−5.51	0.86	−7.35–−3.87
Type AC (None)	1.10	0.09	1.13	−2.07–2.67
Type AC (Window)	4.68	1.54	0.57	0.39–2.74 *
OpenWindow (Yes)	3.73	1.32	0.58	0.21–2.58 *
WaterStorage (Yes)	2.83	1.04	0.53	−0.05–2.10
OtherContainers	1.01	0.01	0.00	0.005–0.02
AP2.1	0.49	−0.71	0.22	−1.16–−0.27 *
Window1	2.12	0.75	0.27	0.23–1.29 *
Door2	1.66	0.51	0.26	0.01–1.09 *
Outdoor female	1.01	0.01	0.00	0.005–0.02

* Variables considered statistically significant.

**Table 7 insects-12-00183-t007:** Main effects statistics for the best fit generalized linear model (NB1) for outdoor female *Ae. aegypti* abundance in South Texas.

Variable	Exp (Estimate)	Estimate	Std. Error	95% CI
(Intercept)		2.27	0.28	1.70–2.81
Vegetation (>51%)	0.38	−0.96	0.30	−1.55–−0.36 *
OpenDoor	0.30	−1.19	0.27	−1.74–−0.66 *
Tires	0.92	−0.08	0.02	−0.12–−0.04 *
Income (>$75k)	0.81	−0.21	0.38	−0.98–0.53
Income ($25–$50k)	5.01	1.61	0.31	0.99–2.23 *
AP2.1	1.65	0.50	0.11	0.27–0.72 *
AP2.2	1.40	0.34	0.15	0.01–0.64 *
Door1	1.23	0.20	0.07	0.01–0.34 *
Door2	0.70	−0.35	0.09	−0.54–−0.17 *

* Variables considered statistically significant.

## Data Availability

The Datasets and R code are available in the Appendix A: R code.

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
