# Peer review of "The Eco-Bio-Social Factors That Modulate Aedes aegypti Abundance in South Texas Border Communities"

_insects, 2021, doi:10.3390/insects12020183_

Round 1

Reviewer 1 Report

The authors describe and analyze the bionomics of Aedes aegypti studying factors influencing its distribution and relative abundance in domestic and peridomestic environment and also how the community members perceive the risk of the vector presence and the prevention measures they implement.

The two-year field study was carried out in an in-depth and adequately structured way, identifying the key points and fundamental phases of the work.

Furthermore the Supplementary Materials are very exhaustive; by these it is possible to deepen and verify all contents described within the manuscript.

The present work is rigorous, detailed and well organized and executed with due respect to scientific methodology and needs only a minor revision.

Very little suggestions are required and listed below:

Lines 65-66: ….Aedes spp. Mosquitoes… already written a few lines above, maybe better replace with “…these species..

Line 76:   better replace with singular form“impact

Lines 86-90: Better to merge the last two sentences “…..diseases they transmit, allowing us to identify…”

Line 145: Caption of figure 2. In the first sentence “Pictorial diagram and field usage of the Center for Disease Control AGO” maybe is better “Pictorial diagram and field usage of the AGO trap (Source: CDC, Spring Star, KING 5 - modified)”

Line 246:However, fewer respondents could identify mosquito larvae...”.  The sentence seems temporally consecutive with the previous one. Better to change in “However, fewer respondents are/were able to identify mosquito larvae…

Line 411: “..things ….” better replace with “factors

Line 494 and following: Uniform the fonts of publication dates in the references section. Some ref. have the date in bold.

Reviewer 2 Report

Line 53-54. Ae. aegypti has been a concern a lot longer than just due to recent epidemics.

Line 58. Change ‘to’ to ‘which prevents’

The results section is largely redundant. The authors use words to describe their results and then have tables reporting the same results. I would recommend picking one or the other.  

Line 423. Please describe what the Sustainable Development Goals are

Reviewer 3 Report

This is a nice paper that highlights some factors influencing the abundance of Aedes aegypti mosquitoes in houses and yards in the Lower Rio Grande Valley, Texas. Interestingly, the paper reports houses with younger children have fewer mosquitoes outside but more inside houses. This is useful information to relay to the community to reduce mosquito contacts around homes. However, there are some shortcomings that could affect the interpretation of the results. One is that only gravid traps were used to collect mosquitoes, which. If the authors had also included host-seeking mosquitoes, the results would more directly relate the results to human-mosquito encounters. The authors should discuss the implications of this sampling technique. Second, it’s unclear why the authors include some social variables in models explaining outdoor mosquito abundance. In particular, including hypotheses in the introduction would clarify the reasoning and expectations for the choices of predictors in the models. It does become more clear in the discussion, but this should be framed earlier in the paper.

Some minor issues are articulated below:

Some grammar issues throughout.

L 89: LGRV

L152: how do you define “household heads”? And why were they the only ones surveyed?

L177: how do the AGO trap affect aegypti abundance? Please elaborate

L182: technically, it’s the error that follow a Poisson distribution

L183: unclear what you compared; models with different error distributions or the error distributions themselves? It seems you are explaining how you determined which error distribution to use; it would be useful to include that statement in this paragraph

L211 and vicinity: explain tradeoffs of grouping data in this way (e.g. loss of information, explanatory resolution)

Table 2: how was “messy yard” determined in a standardized and unbiased way?

L 268: check wording/grammar

L269: change “in their property” to “on their property”

L281-284: how much variation was captured in AP2?

L335: how did you determine the income categories? It seems like 25-75K is a pretty wide range, particularly compared to <25K

L340: please explain why you use house characteristics (e.g. number of doors), in addition to AP1 and AP2 in the models for outdoor abundance. It would be helpful to have specific hypotheses laid out for environmental versus social variables (e.g. why do you expect survey data to relate to outdoor abundance?)
